# Mycotoxin Deoxynivalenol Has Different Impacts on Intestinal Barrier and Stem Cells by Its Route of Exposure

**DOI:** 10.3390/toxins12100610

**Published:** 2020-09-24

**Authors:** Hikaru Hanyu, Yuki Yokoi, Kiminori Nakamura, Tokiyoshi Ayabe, Keisuke Tanaka, Kinuko Uno, Katsuhiro Miyajima, Yuki Saito, Ken Iwatsuki, Makoto Shimizu, Miki Tadaishi, Kazuo Kobayashi-Hattori

**Affiliations:** 1Department of Nutritional Science, Faculty of Applied Bioscience, Tokyo University of Agriculture, 1-1-1 Sakuragaoka, Setagaya-ku, Tokyo 156-8502, Japan; hikaruhanyu@gmail.com (H.H.); yuki1920s@gmail.com (Y.S.); ms205346@nodai.ac.jp (M.S.); mt205315@nodai.ac.jp (M.T.); 2Department of Cell Biological Science, Faculty of Advanced Life Science, Hokkaido University, Kita-21, Nishi-11, Kita-ku, Sapporo, Hokkaido 001-0021, Japan; y-yokoi-b-s@eis.hokudai.ac.jp (Y.Y.); kiminori@sci.hokudai.ac.jp (K.N.); ayabe@sci.hokudai.ac.jp (T.A.); 3NODAI Genome Research Center, Tokyo University of Agriculture, 1-1-1 Sakuragaoka, Setagaya-ku, Tokyo 156-8502, Japan; kt205453@nodai.ac.jp; 4Department of Nutritional Science and Food Safety, Faculty of Applied Bioscience, Tokyo University of Agriculture, 1-1-1 Sakuragaoka, Setagaya-ku, Tokyo 156-8502, Japan; 45219001@nodai.ac.jp (K.U.); km206186@nodai.ac.jp (K.M.); ki204886@nodai.ac.jp (K.I.)

**Keywords:** deoxynivalenol, intestinal barrier, intestinal stem cells, enteroids, basolateral exposure, luminal exposure, mycotoxin, organoids, microinjection

## Abstract

The different effects of deoxynivalenol (DON) on intestinal barrier and stem cells by its route of exposure remain less known. We explored the toxic effects of DON on intestinal barrier functions and stem cells after DON microinjection (luminal exposure) or addition to a culture medium (basolateral exposure) using three-dimensional mouse intestinal organoids (enteroids). The influx test using fluorescein-labeled dextran showed that basolateral DON exposure (1 micromolar (µM) disrupted intestinal barrier functions in enteroids compared with luminal DON exposure at the same concentration. Moreover, an immunofluorescence experiment of intestinal epithelial proteins, such as E-cadherin, claudin, zonula occludens-1 (ZO-1), and occludin, exhibited that only basolateral DON exposure broke down intestinal epithelial integrity. A time-lapse analysis using enteroids from leucine-rich repeat-containing G-protein-coupled receptor 5 (Lgr5)-enhanced green fluorescence protein (EGFP) transgenic mice and 5-ethynyl-2-deoxyuridine (EdU) assay indicated that only the basolateral DON exposure, but not luminal DON exposure, suppressed Lgr5^+^ stem cell count and proliferative cell ratio, respectively. These results revealed that basolateral DON exposure has larger impacts on intestinal barrier function and stem cells than luminal DON exposure. This is the first report that DON had different impacts on intestinal stem cells depending on the administration route. In addition, RNA sequencing analysis showed different expression of genes among enteroids after basolateral and luminal DON exposure.

## 1. Introduction

Deoxynivalenol (DON) is a mycotoxin produced by the Fusarium species, commonly contaminating cereal-based foods, such as wheat, rye, barley, oats, and corn [1,2]. DON is resistant to milling, processing, and heating [3] and, therefore, readily enters the food chain, causing food poisoning. Several outbreaks of acute human gastrointestinal disorders, including nausea, vomiting, diarrhea, and gastrointestinal discomfort, have been reported in Asia [4,5].

The gastrointestinal tract is the first physiologic barrier against food contaminants, as well as the first target for toxicants. DON reportedly affects the small intestine, and, most notably, its negative effects on intestinal barrier functions were demonstrated in a previous work [6]. Intestinal barrier functions are primarily formed by intestinal epithelial cells connected by tight junction proteins, forming an anastomosing network that seals adjacent epithelial cells near the luminal surface [7,8]. Tight junctions are composed of (1) transmembrane proteins, including occludins and claudins, which form a linear barrier at the luminal lateral cell membranes, and (2) peripheral membrane proteins, such as zonula occludens (ZO) proteins, which link transmembrane tight junction proteins together [9]. DON also reportedly affects tight junction proteins by decreasing claudin proteins after DON exposure in vitro [10].

Interestingly, DON is rapidly absorbed while passing through the stomach and the proximal small intestine [11]. This report suggests that DON affects the intestinal cells not only from the luminal side but also from the basolateral side through the blood stream. A few scientists previously explored the different effects of DON from both the luminal and basolateral routes using porcine intestinal cell lines [12,13,14] or human epithelial cell lines [15]. They reported that basolateral DON exposure has greater impacts on intestinal barrier functions, cell proliferation, and interleukin-8 secretion than its luminal exposure. However, the intestinal cell lines used in previous studies were composed of enterocyte monolayers without intestinal crypt-villus structures and native intestinal proliferation and differentiation functions. Additionally, these cell lines lacked differentiated cells, such as goblet cells, enteroendocrine cells, Paneth cells, and tuft cells, because intestinal stem cells were absent.

Recently, intestinal organoids (enteroids) are widely used as in vitro models that mimic most features of intestinal epithelium in vivo. They consist of intestinal stem cells, as well as most types of differentiated cells derived from stem cells, and are easily generated from isolated crypts as previously shown [16,17]. Establishing enteroid cultures was a breakthrough method in evaluating the effects of substances on intestinal stem cells in vitro. Until recently, little information was available on the DON toxicity on intestinal stem cells. Only a few enteroid-based studies showed that adding DON to the enteroid culture medium suppressed intestinal cell proliferation and decreased the protein level of leucine-rich repeat-containing G-protein-coupled receptor 5 (Lgr5), a stem cell marker in the small intestine [18,19]. However, their studies only reflected the effects of basolateral DON exposure on enteroids due to their reversed basolateral-out structure, in contrast to the native intestine. Therefore, the effects of luminal DON exposure on intestinal stem cells remain unclear.

Our goal was to investigate the impacts of luminal or basolateral DON exposure on the small intestine using enteroids. In the present study, we focused on the harmful effects of DON on intestinal barrier functions, as well as intestinal stem cells, after luminally or basolaterally exposed to DON.

## 2. Results

### 2.1. Microinjection into Enteroids as Luminal Exposure

Enteroids have a three-dimensional structure that consists of a central lumen lined by villus-like epithelium and several crypt-like domains (Figure 1A). Strong actin staining of microvilli brush borders showed that luminal enteroid surfaces face inward (Figure 1A). Using enteroids as an in vitro model to test the effects of luminal toxin exposure is difficult because of their closed structures. Thus, we used microinjection technique to deliver substances to the enteroid lumen as previously described [20] (see Appendix A). Between before and after microinjection (Figure 1B, Appendix A), the microinjection solution concentration was diluted by the enteroid lumen volume. The average volume of the enteroid lumen and the average sample volume injected with the microneedle were measured to compensate for this dilution and adjust the substance concentration that was microinjected into the lumen (Appendix A). These results showed that substance concentration was diluted 28.3 times through microinjection into the enteroid lumen (Appendix A). We also did not detect any leakage from the enteroid lumen after 10 kDa Alexa Fluor^®^ dextran microinjection into the enteroids (Figure 1C).

### 2.2. Basolateral DON Exposure Disrupted Intestinal Barrier Functions

For all experiments, we set a 1- micromolar (µM) DON concentration from the results of the 3-(4,5-dimethylthiazol-2-yl)-2,5-diphenyltetrazolium bromide (MTT) viability assay [21] to prevent potent toxic side effects (Appendix A) (see Appendix A). Alexa Fluor^®^ dextran (molecular weight (MW): 10 kDa) was added to the enteroid culture medium after luminal or basolateral DON (final concentration (conc.) 1 µM) exposures to test its toxic effects on intestinal barrier functions. For the luminal DON exposures, we injected 28.3 µM DON solution, since the substance concentration was diluted 28.3 times through microinjection into the enteroid lumen as shown in Figure 1, Appendix A. The 96-h time-lapse live imaging indicated that the basolateral-to-apical leakage of fluorescent dextran occurred in enteroids after basolateral DON exposure (Figure 2A). In addition, the fluorescence intensity ratio was increased in enteroids only after basolateral DON exposure (Figure 2B), confirming that the fluorescence intensity ratio of enteroids basolaterally treated with DON at 96 h was significantly higher compared with that of other treatments (Figure 2C).

### 2.3. Basolateral DON Exposure Broke down Intestinal Epithelial Integrity

Immunofluorescence of intestinal epithelial proteins (E-cadherin, claudin, and occludin) was performed in enteroids at 72 h after luminal and basolateral DON exposures. Microinjection had no effect on the structure of the PBS-injected enteroids in comparison with control (untreated) enteroids (Figure 3). Similarly, enteroids luminally exposed to DON did not show obvious effects, compared with the control enteroids (Figure 3). In contrast, E-cadherin, the core transmembrane protein of the adherens’ junction, was disrupted only after basolateral DON exposure (Figure 3). Likewise, ZO-1, claudin-2, and occludin, important tight junction proteins, were broken down in enteroids that were basolaterally exposed to DON (Figure 3).

### 2.4. Basolateral DON Exposure Suppressed Intestinal Stem Cells

The 24-h time-lapse live imaging of enteroids derived from Lgr5- enhanced green fluorescence protein (EGFP) transgenic mice revealed that basolateral DON treatment reduced Lgr5-EGFP^+^ cells, in comparison with other treatment groups (Figure 4A, Appendix A). Furthermore, the ratio of Lgr5^+^ stem cell number was significantly decreased in enteroids after basolateral DON exposure (Figure 4C). In contrast, no change was observed in the ratio of Lgr5^+^ stem cell number in enteroids luminally exposed to DON (Figure 4C). Next, the 5-ethynyl-2-deoxyuridine (EdU) assay was performed in enteroids with luminal or basolateral DON exposure for 24 h to visualize the red-stained proliferated cells (Figure 4B). The EdU^+^ cell number ratio was significantly lower in enteroids with basolateral DON exposure than in other treatment groups (Figure 4D). There was no confirmed effect of microinjection itself on intestinal stem cells or intestinal cell proliferation between the control enteroids and the PBS-microinjected enteroids (Figure 4A–D).

### 2.5. Oral Administration of DON to Mice Suppressed Intestinal Stem Cell Viability

Next, we investigated whether the effect of DON on intestinal stem cells confirmed in our enteroid model is observed in vivo. C57/BL6 mice were orally administrated with DON at a dose of 50 mg/kg body weight after fasting overnight, and the intestinal crypt was isolated after 24 h of DON exposure (Figure 5A). The enteroids prepared from the crypts were cultured for four days before evaluation (Figure 5A). Enteroid-forming efficiency, broadly used to assess stem cell function [18,19,22,23], was significantly decreased in enteroids from mice orally administrated with DON (Figure 5B,C). Furthermore, intestinal cell proliferation was significantly suppressed in enteroids from the mice after oral DON administration (Figure 5D,E).

### 2.6. RNA Sequencing (RNA-Seq) Analysis Revealed the Different Expressions of Genes between Basolateral and Luminal DON Exposures

RNA-seq analysis was conducted among control enteroids, PBS-microinjected enteroids, and enteroids after luminal or basolateral DON exposures. The heat map among 888 differentially expressed genes (DEGs) from enteroids after the basolateral DON exposure clearly differed from that of the other treatment groups (Figure 6A). The greater number of up- or downregulated transcripts, in comparison with the control, was changed in enteroids after basolateral DON exposure (total: 532 DEGs) than in those after luminal DON exposure (total: 70 DEGs) (Figure 6B). The database for annotation, visualization and integrated discovery (DAVID) gene set enrichment analysis revealed that the focal adhesion pathway (mmu04510) was the top-ranked pathway, when comparing between enteroids after luminal DON exposure and after basolateral DON exposure (*p* = 0.001796) (Appendix A). Additionally, quantitative polymerase chain reaction (qPCR) analysis was conducted to confirm the changes of key genes in the focal adhesion pathway, resulting in gene expression trends similar to the RNA-seq results (Figure 3 and Appendix A) (see Appendix A). Different gene expression changes were also observed in the mitogen-activated protein kinases (MAPK) signaling pathway (mmu04010) and Wnt signaling pathway (mmu04310) (Appendix A), and similar gene expression trends were confirmed through qPCR analysis (Appendix A).

## 3. Discussion

Several studies have reported that DON disrupts normal intestinal barrier functions [6,10,24,25,26,27]. However, only a few studies reported the effects of luminal or basolateral DON exposure on the intestine using intestinal cell lines [12,14]. Thus, we first addressed if luminal or basolateral DON exposure affects intestinal barrier functions using our enteroid model, which mimics most features of the native intestine. Recently, Yokoi et al. [20] devised a novel microinjection method that quantitatively luminally delivers substances to enteroids. This method allowed us to compare between the effects of luminal and basolateral DON treatment at equal concentrations (Figure 1). Our experiment using fluorescent dextran (Figure 2) demonstrated that basolateral DON exposure had more impact on intestinal barrier functions than luminal DON exposure, agreeing with previous reports using intestinal cell lines [12,14]. We hypothesized that basolateral DON exposure may disrupt the intestinal barrier by affecting the intestinal epithelial structure, since the intestinal barrier is assisted by the intestinal epithelial proteins involved in the adherens junction or tight junction. Immunofluorescence of intestinal epithelial proteins (E-cadherin, claudin, and occludin) in enteroids (Figure 3) revealed that only basolateral DON exposure broke down intestinal epithelial integrity. Taken together, basolateral DON exposure had more negative effects on intestinal epithelial proteins that have important roles in intestinal barrier functions, compared with luminal DON exposure. Moreover, our enteroid model enabled us to visually assess intestinal barrier functions, which has more similarity to native intestinal epithelium than 2-D cell line models used in previous studies.

The molecular mechanism by which basolateral DON exposure affected intestinal barrier functions is reportedly due to the occurrence where DON decreases tight junction proteins by activating the MAPK pathway [24,26]. In the present study, the basolateral DON exposure did not significantly change extracellular signal-regulated kinase (ERK), p38 mitogen activated protein kinase (p38), or c-jun N-terminal kinase (JNK) gene expression in enteroids in comparison with the control (Appendix A). Based on our RNA-seq results, MAPK signaling activation possibly caused the disruption of intestinal barrier functions in enteroids basolaterally exposed to DON in the present study, which is similar to previous report [24,26].

Although we demonstrated that the basolateral DON has lager impacts on intestinal barrier function in our enteroid model, the same as previous reports, our interest was how luminal or basolateral DON affects intestinal stem cells. The effects of DON exposure on intestinal stem cells have not been tested in vitro due to the lack of a model that contains intestinal stem cells, although the shortened villi observed in DON-administrated mice implied that it affects intestinal cell proliferation and/or intestinal stem cells [6,28,29,30]. Subsequently, establishing a protocol to generate enteroids was the breakthrough in studying intestinal stem cells in vitro [17]. Using this model, Li et al. [22] demonstrated that basolateral DON exposure (adding DON to a porcine enteroid culture medium) inhibited intestinal stem cell activity through the Wnt/β-catenin pathway. Thus, we further explored the effects of luminal or basolateral DON exposure on intestinal stem cells. Our outcomes (Figure 4) demonstrated that basolateral DON exposure greatly affected intestinal stem cells than did luminal DON exposure at equal concentrations. Next, we examined the effects of DON on stem cells in native small intestine of mice with oral administration (50 mg/kg BW DON) (Figure 5). Our purpose of conducting the in vivo experiment was to know whether DON has similar toxic effects in vivo to those observed on stem cells of enteroids. Thus, we used 50 mg/kg BW of DON as the dose, based on our preliminary experiments. Although we did not measure DON concentration in the blood, Pestka et al. [31] showed that DON level in the blood was about 0.4 μg/mL at 24 h after a single oral exposure of 25 mg/kg BW of DON. In the case of 50 mg/kg BW of DON, DON concentration in the blood is expected to be about 0.8 μg/mL (2.7 μM), which is almost comparable to the concentration that was used in our in vitro experiment (1 μM DON). Enteroids derived from mice orally administrated with DON showed reduced forming efficiency and proliferative activity in comparison with the control enteroids (Figure 5), which is consistent with our in vitro results, showing that the in vitro enteroid model can be a powerful tool to test the effects of toxins on intestinal stem cells.

We assumed that DON has greater impacts on intestinal stem cells than on differentiated cells, although the present study did not investigate DON’s effects on differentiated cells. Proliferating cells, such as intestinal stem cells, are frequently expected to be highly susceptible to DON, since DON has an ability to inhibit protein synthesis [32,33]. Indeed, the DON concentration (1 µM) used in our enteroid experiment, which negatively affected intestinal stem cells, was lower than that in previous works using cell lines [32,33]. The RNA-seq analysis in our study revealed that only the basolateral DON exposure downregulated the gene expression of cyclin D, a target gene of the Wnt signaling pathway (Appendix A). This is supported by a previous study that reported DON to have an effect on stem cells through the Wnt/β-catenin pathway [22]. However, based on our RNA-seq results, the gene expressions of Wnt or frizzled-class receptors were upregulated only after basolateral DON exposure (Appendix A). This may be the negative feedback regulation to compensate for the suppression of Wnt/β-catenin pathway caused by DON exposure. Taken together, the high susceptibility to DON and the downregulation of the Wnt/β-catenin pathway may be attributed to the decreased stem cell number.

The question of why basolateral DON exposure has more impacts on the intestine than luminal DON exposure at the same concentration still remains. There are three possible answers: Firstly, basolateral DON could enter the cells potentially faster or more easily, compared with apical entry, because P-glycoprotein (P-gp) in the apical membrane exports drugs out of the cells, as previously discussed [14]. Certainly, Ivanova et al. have reported that DON is exported by P-gp, causing it to be less cytotoxic in P-gp-overexpressing cells [34]. Secondly, our RNA-seq analysis showed that several gene expressions, including extracellular matrix (ECM) in the focal adhesion pathway (mmu04510), were upregulated not by luminal DON exposure, but by basolateral exposure (Figure 7). This result implies that basolateral DON exposure caused extracellular damage to the focal adhesion pathway, subsequently inducing feedback regulation. Gene expression changes in the focal adhesion pathway suggest that the basolateral DON is likely to enter the intestinal epithelium, leading to greater impacts than luminal DON. Finally, the luminal mucus layer of enteroids may inhibit DON entry from the luminal side, thereby lowering the toxic effects of luminal DON exposure. Enteroids stained with periodic acid-Schiff showed that there were goblet cells and secreted mucus substances that covered the enteroid lumen surface (Appendix A) (see Appendix A). However, in the present study, the detailed, involved mechanisms still remain unclear. Further studies on these mechanisms are required.

In conclusion, our study revealed that basolateral DON exposure was more toxic than luminal DON exposure in terms of intestinal barrier functions and stem cells. In particular, the present study is the first to show that the basolateral DON exposure had greater negative effects on intestinal stem cells than luminal DON exposure. Individually clarifying the effects of DON on the intestine both from the luminal and basolateral sides is important, since DON is rapidly and nearly completely absorbed in the stomach and the proximal small intestine [11]. Our findings provide useful information in accurately understanding DON toxicity on intestinal barrier functions and stem cells. Additionally, our enteroid model is animal-friendly and may be technically useful for evaluating the effects of toxins on intestinal stem cells and intestinal epithelium functions, because this enteroid-based in vitro model reduces the experimental period, cost, and number of animals used, compared with in vivo models.

## 4. Materials and Methods

### 4.1. Experimental Animals

C57BL/6J (wild-type; WT) mice (CLEA Japan, Inc., Tokyo, Japan) and Lgr5-EGFP-internal ribosome entry site (IRES)-CreERT2 (Lgr5-EGFP) mice (C57BL/6 background; The Jackson Laboratory, Bar Harbor, ME, USA) were fed with standard laboratory rodent food (MF; Oriental Yeast Co., Ltd., Tokyo, Japan), with free access to water. Male and female mice were used for experimentation at 2–4 months of age. The Lgr5-EGFP mice were used to visualize intestinal stem cells in enteroids. For in vitro experiments, we prepared enteroids from three animals (approximately half male and female mice) in each experiment. For oral DON administration, the WT mice were fasted overnight with free access to water and then orally treated with DON at a dose of 50 mg/kg body weight (two male mice for the control and three male mice for the DON-treated group). Mice were sacrificed for crypt isolation at 24 h after oral DON administration. All animal experiments adhered to the guidelines for maintaining and handling of experimental animals established by the Tokyo University of Agriculture Ethics Committee. Approval code and date of the ethics committee: 2019040 (27 September 2019) and 290015 (28 September 2017)

### 4.2. Generation of Mouse Enteroids

Enteroids from the mouse jejunum were generated as previously described [16,17]. Briefly, the isolated mouse jejunum was incubated with rocking at 4 °C for 30 min in 2.5 mM ethylenediaminetetraacetic acid (EDTA) in Dulbecco’s phosphate-buffered saline (DPBS; Nacalai Tesque, Inc., Kyoto, Japan) without calcium or magnesium. The tissue was then placed in an ice-cold dissociation buffer (43.3 mM sucrose (FUJIFILM Wako Pure Chemical Corporation, Osaka, Japan) and 54.9 mM D-sorbitol (Nacalai Tesque, Inc., Kyoto, Japan) in DPBS) and was forcefully shaken by hand to dissociate individual crypts. The dissociated crypts were centrifuged at 400× *g* for 4 min at 4 °C, and the pellet was resuspended in 60% Matrigel (Corning Incorporated, Corning, NY, USA) in ice-cold DPBS. The crypts suspended in Matrigel were added to a 24-well cell culture plate (NIPPON Genetics Co., Ltd., Tokyo, Japan). After Matrigel polymerization at 37 °C, enteroid culture medium (advanced Dulbecco’s modified Eagle medium/Ham’s F-12 (DMEM/F-12; Thermo Fisher Scientific Inc., Waltham, MA, USA) supplemented with 2 mM L-glutamine (Nacalai Tesque, Inc., Kyoto, Japan), 10 mM 2-[4-(2-hydroxyethyl)piperazin-1-yl]ethanesulfonic acid (HEPES; Nacalai Tesque, Inc., Kyoto, Japan), 1× penicillin–streptomycin (Nacalai Tesque, Inc., Kyoto, Japan), 1× B27 supplements (Thermo Fisher Scientific Inc., Waltham, MA, USA), 1 mM N-acetyl-L-cysteine (Sigma–Aldrich Co., LLC, St. Louis, MO, USA), 10% R-spondin conditioned medium, 5% Noggin conditioned medium, and 50 ng/mL epidermal growth factor (EGF; PeproTech Inc., Rocky Hill, NJ, USA)) was added to the wells. Enteroids were cultured in a 5% CO_2_ incubator at 37 °C until day 3. The enteroid culture medium was replaced every two days. For the experiment, enteroids were collected and resuspended in ice-cold 100% Matrigel and transferred into a 35-mm cell imaging dish (Eppendorf AG, Hamburg, Germany). After Matrigel polymerization at 37 °C, the enteroid culture medium was added into each dish. The enteroids were prepared from the control mice or mice orally treated with DON using the aforementioned protocol for assessing the enteroids from orally DON-treated mice. In this experiment, we used two C57/BL6 male mice for the control and three male mice for the DON-treated group. Enteroid-forming efficiency was calculated as the ratio of the number of enteroids at day 4 to the number of crypts seeded at day 0. To calculate enteroid-forming efficiency, crypts in four wells per one mouse, that is, 1.5–6.3 × 10^2^ crypts/well (eight wells for the control and 12 wells for the DON-treated group; *n* = 8–12) were counted at day 0. After four days, grown enteroids were counted in the same wells of 12-well cell culture plate.

### 4.3. DON Exposure of Enteroids

For basolateral DON exposure, DON (final conc. 1 µM; Sigma–Aldrich Co. LLC, St. Louis, MO, USA) was added to the enteroid culture medium. In contrast, luminal DON exposure was conducted through DON microinjection (28.3 µM in DPBS) into enteroids. Enteroid lumen microinjection and volume measurement of enteroid lumen and microinjection were done as previously described [20]. The final concentration was equivalent to 1 µM in microinjected enteroids (Appendix A). Briefly, the tip of the needle (Femtotips II; Eppendorf AG, Hamburg, Germany) was broken off and its inner diameter was adjusted to 3–4 µm. The 28.3-µM DON solution (final conc. 1 µM) was loaded into the needle using a microloader tip (Eppendorf AG, Hamburg, Germany). Microinjection was performed using a microinjector (FemtoJet^®^ 4i; Eppendorf AG) and manipulators (MM-92 and MMO-202ND; Narishige Scientific Instrument Lab, Tokyo, Japan) under a microscope (Olympus Co., Tokyo, Japan). Enteroid lumen was stained with 1 µM rhodamine123 (Rh123) for 3 h (Appendix A) to measure its volume. Likewise, DPBS was microinjected into the enteroids to test the effects of microinjection itself (described as “PBS injection” in the figures). For the experiment to test dextran permeability, DPBS was microinjected into the enteroids in the basolateral DON exposure group before exposure in view of the effects of microinjection itself.

### 4.4. Long-Term Live Imaging of Enteroids

Live imaging was performed on the enteroid culture medium at 37 °C under 5% CO_2_ (incubation chamber conditions) on a confocal microscope (Fv10i; Olympus Co., Tokyo, Japan) and imaged with a 40× objective lens. To assess intestinal barrier functions, 10 μM Alexa Fluor^®^ 647–10K Dextran (excitation: 633 nm, emission: 650–700 nm; Thermo Fisher Scientific Inc., Waltham, MA, USA) was added to the enteroid culture medium for 96 h. To test the effects of DON on intestinal stem cells, the fluorescence of Lgr5-EGFP in the enteroids was monitored for 24 h (excitation: 488 nm, emission: 500–550 nm). Five to seven enteroids in each group were imaged for each experiment using one mouse.

### 4.5. Whole-Mount Immunostaining

Enteroids with or without DON exposure for 72 h were fixed with 4% cold paraformaldehyde (Nacalai Tesque, Inc., Kyoto, Japan) for 30 min at room temperature. The enteroids were then blocked with 5% goat serum (Jackson ImmunoResearch Laboratories, Inc, West Grove, PA, USA) for 1 h at room temperature before incubating with ZO-1 (1:100; Cat #33-9100; Thermo Fisher Scientific Inc., Waltham, MA, USA), E-cadherin (1:400; Cat #AF748-SP; R&D systems, Inc., Minneapolis, MN, USA), claudin-2 (1:250; Cat #351-6100; Thermo Fisher Scientific Inc., Waltham, MA, USA), or occludin (1:250; Cat #33-1500; Thermo Fisher Scientific Inc., Waltham, MA, USA) antibodies overnight at 4 °C. The enteroids were incubated with Alexa Flour^®^ 488, 647, or 555 (1:1000; Thermo Fisher Scientific Inc., Waltham, MA, USA) secondary antibodies overnight at 4 °C after washing all day with 0.1% Triton-X in PBS. Additionally, the enteroids were incubated with 10 μg/mL Hoechst 33342 (Thermo Fisher Scientific Inc., Waltham, MA, USA) for 20 min. Finally, the enteroids (five to seven enteroids in each group per one mouse) were imaged using a confocal microscope (Fv10i; Olympus Co., Tokyo, Japan).

### 4.6. EdU Assay

Enteroids with or without the DON exposure for 24 h were incubated with 10 μM EdU (Thermo Fisher Scientific Inc., Waltham, MA, USA) for 3 h. Then, 10 μg/mL Hoechst 33342 (Thermo Fisher Scientific Inc., Waltham, MA, USA) was added to the enteroid culture medium for 30 min prior to fixation with 4% cold paraformaldehyde (Nacalai Tesque, Inc., Kyoto, Japan) for 30 min at room temperature. The enteroids were stained using a Click-iT™ EdU Cell Proliferation Kit for Imaging (Cat# C10339; Thermo Fisher Scientific Inc., Waltham, MA, USA), according to the manufacturer’s protocol. Briefly, the enteroids were permeabilized with 0.5% Triton X-100 in PBS for 20 min and incubated with Click-iT detection cocktail for 30 min to detect EdU^+^ cells. The enteroids (5–7 enteroids in each group per one mouse) were imaged using a confocal microscope (Fv10i; Olympus Co., Tokyo, Japan). EdU assay was conducted, similarly to the aforementioned protocol, to investigate enteroids from mice orally treated with DON.

### 4.7. Image Analysis

For the dextran permeability experiment, Alexa Fluor^®^ 647-10K Dextran intensity in enteroid lumen was measured every hour until 96 h. For quantification, data were collected from 17 enteroids and 18 organoids in the PBS injection or the luminal DON exposure groups and in the basolateral DON exposure group, respectively, in total from three different mice (Figure 2B,C; *n* = 17–18). The fluorescence intensity ratio was calculated as follows: The fluorescence intensity in the enteroid lumen at various timepoints was normalized with that at 0 h (set to 1), and the values were relatively described as the fluorescence intensity ratio.

For the live imaging of Lgr5-EGFP enteroids, Lgr5^+^ cells in the enteroids were counted every hour until 24 h. Data were collected from eight enteroids in the control group, 12 enteroids in the basolateral DON exposure group, 13 enteroids in the PBS injection group, and 16 enteroids in the luminal DON exposure group in total from three different mice **(**Figure 4C; *n* = 8–12). The Lgr5^+^ stem cell number was normalized to the number of Lgr5^+^ cell at 0 h after each treatment, which was relatively expressed as the Lgr5^+^ cell number ratio.

For the EdU proliferation assay, the percentage of the EdU^+^ cell number to total cell number was calculated and represented as EdU/total cells (%). Data were collected from 16 enteroids in the control group, 19 enteroids in the PBS injection group, 20 enteroids in the luminal DON exposure group, and 22 enteroids in the basolateral DON exposure group in total from three different mice (Figure 4D; *n* = 16–22). Similarly, data were collected from 12 enteroids in control group and 18 enteroids in DON-treated group in total from 2–3 different mouse (Figure 5E; *n* = 12–18).

### 4.8. RNA-Seq Analysis

The total RNA was isolated from the control or DON-exposed enteroids using ISOGEN II (Nippon Gene Co., Ltd., Tokyo, Japan). The isolated total RNA was treated with DNase to eliminate DNA contamination. The RNA library was prepared by TruSeq^®^ Stranded mRNA Library Prep. Kit (Illumina, Inc., San Diego, CA, USA), according to the manufacturer’s protocol. The cDNA was amplified, and the obtained cDNA pool was subjected to high-throughput sequencing using the NovaSeq 6000 system. Raw read data quality was checked using the FastQC v0.11.7 software (http://www.bioinformatics.babraham.ac.uk/projects/fastqc/). Based on the scores, each read was cleaned through trimming adapter sequence and filtering low-quality sequence using the Trimmomatic 0.38 software (http://www.usadellab.org/cms/?page=trimmomatic). The cleaned read data were aligned using the HISAT2 software (https://daehwankimlab.github.io/hisat2/) version 2.1.0. Gene expression value was quantified using StringTie version 1.3.4d (https://ccb.jhu.edu/software/stringtie/). The statistical test for comparing between two groups was performed using the R package, edgeR (https://bioconductor.org/packages/release/bioc/html/edgeR.html). Significant DEGs were identified based on the conditions of |fold change| > = 2 and raw *p*-value < 0.05 in at least one comparison group. A functional annotation and gene ontology enrichment analysis were performed based on 539 DEGs between enteroid groups after luminal versus basolateral DON exposures using DAVID Bioinformatics Resources (https://david.ncifcrf.gov/).

### 4.9. Statistical Analysis

All values were reported from representative experiments as the mean ± standard error of the mean (SEM) from multiple experiments using each enteroid derived from at least three animals. We determined the statistical significance using unpaired Student’s *t*-test, or one-way ANOVA with Tukey′s multiple comparison post hoc test. A *p*-value <0.05 was considered significant.

## Figures and Tables

**Figure 1 toxins-12-00610-f001:**
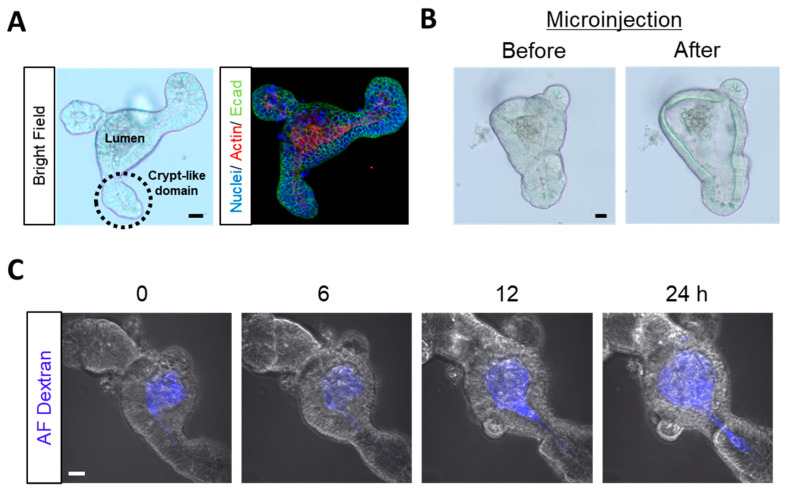
Microinjection into enteroids as luminal exposure. (**A**) Representative images of mice enteroids in a bright field (left) and immunofluorescence (right; blue: Nuclei, red: Actin, and green: E-cadherin). (**B**) Representative images of enteroids before (left) and after (right) microinjection. (**C**) Representative confocal images of enteroids microinjected with Alexa Fluor^®^ (AF) dextran (blue). Scale bars: 20 µm.

**Figure 2 toxins-12-00610-f002:**
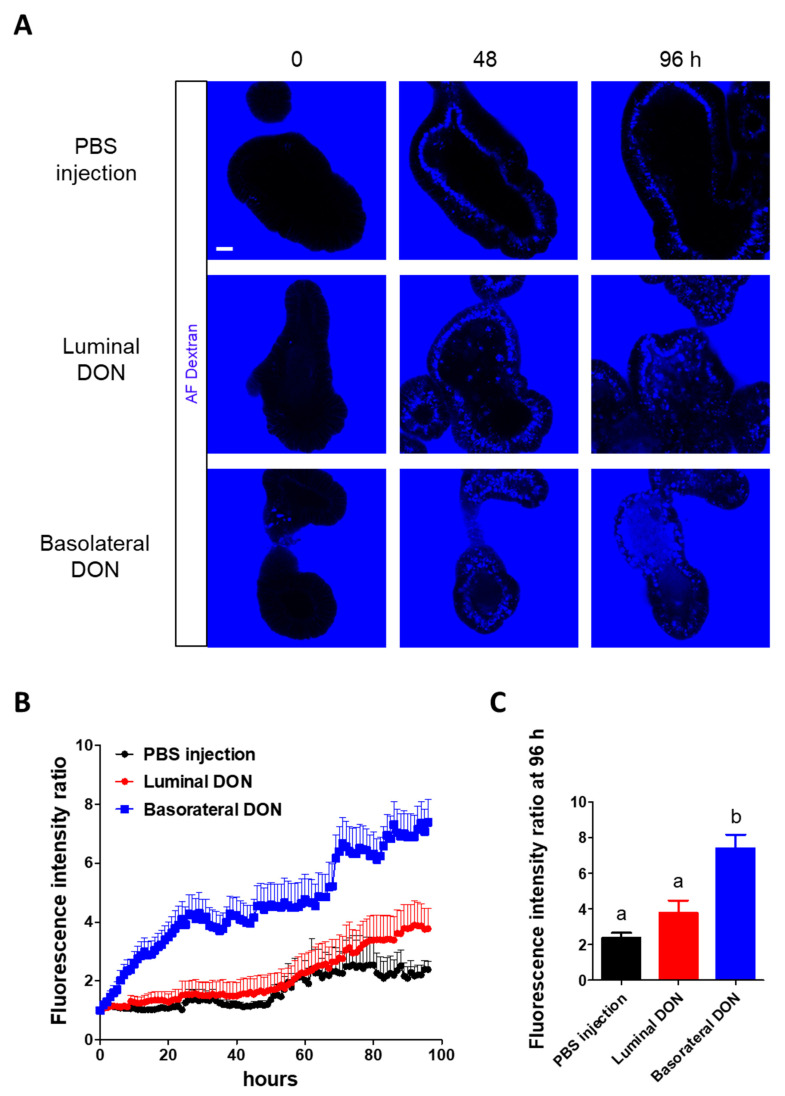
Effects of basolateral or luminal DON exposures on intestinal barrier functions. (**A**) Representative confocal images of enteroids with fluorescent dextran (blue) at 0, 48, and 96 h after treatments (phosphate-buffered saline (PBS) injection, 1-µM luminal DON exposure, or 1-µM basolateral DON exposure). Scale bar: 10 µm. (**B**) The time course of fluorescence intensity ratio of enteroids with fluorescent dextran. Mean ± standard error of the mean (SEM), *n* = 17–18. (**C**) Fluorescence intensity ratio of enteroids (black: Control, red: Luminal DON exposure, and blue: Basolateral DON exposure) at 96 h after treatments. Data were taken from Figure 2B. Mean ± SEM, *n* = 17–18. Different lowercase letters indicate significant differences (*p* < 0.05; Tukey′s post hoc test).

**Figure 3 toxins-12-00610-f003:**
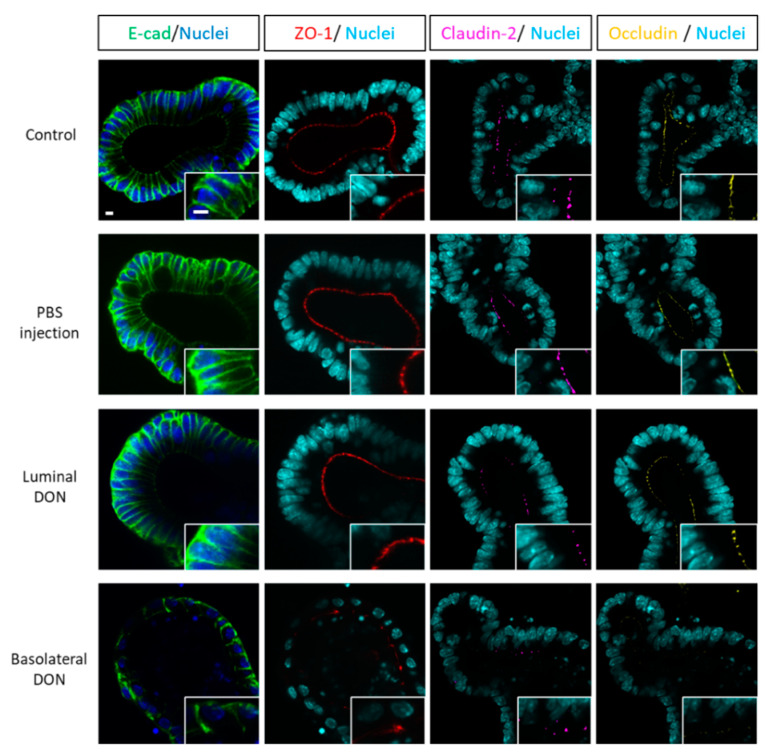
Effects of basolateral or luminal DON exposures on intestinal epithelial integrity. Representative confocal images of enteroids at 48 h after treatments (control, PBS injection, 1-µM luminal DON exposure, or 1-µM basolateral DON exposure). Immunofluorescence shows E-cadherin (green), ZO-1 (red), claudin-2 (pink), occludin (yellow), and nuclei (blue or sky blue). Scale bars: 10 µm.

**Figure 4 toxins-12-00610-f004:**
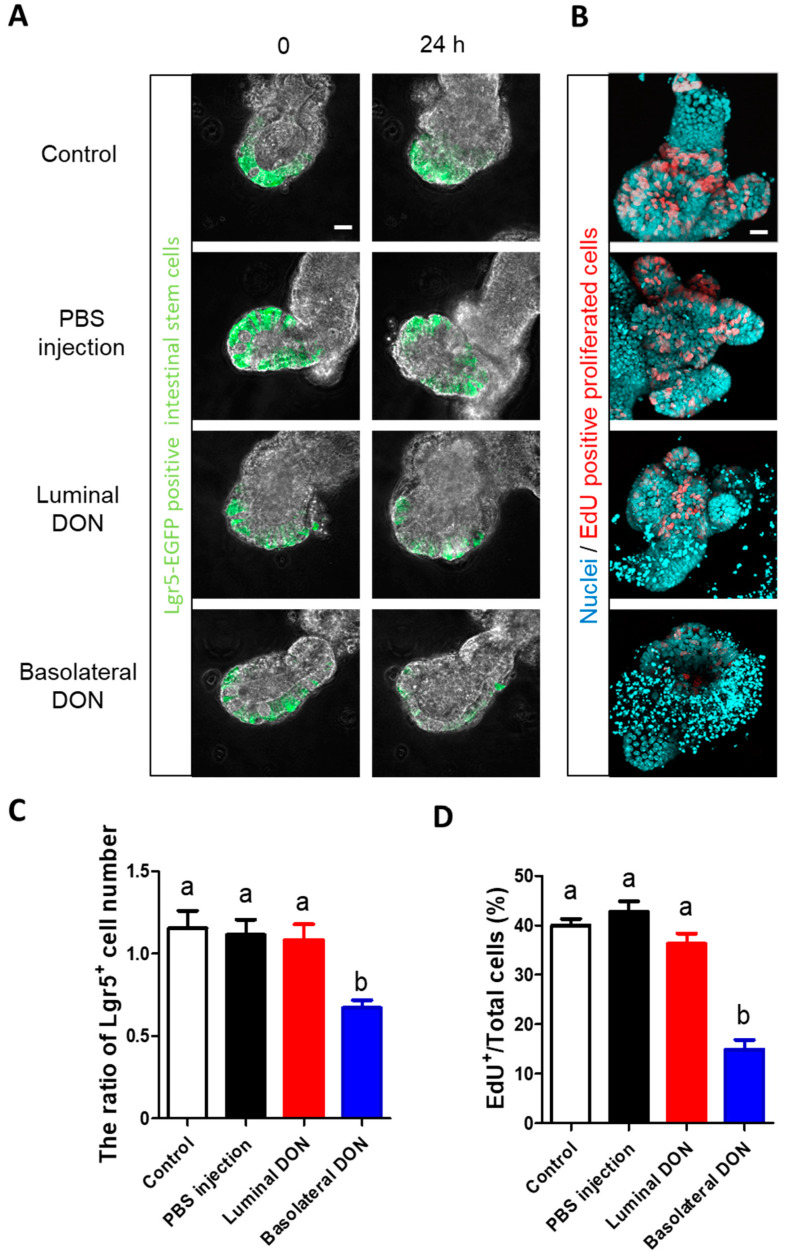
Effects of basolateral or luminal exposures of DON on intestinal stem cells. (**A**) Representative confocal images of Lgr5-enhanced green fluorescence protein (EGFP) enteroids at 0 or 24 h after treatments (control, PBS injection, 1-µM luminal DON exposure, or 1-µM basolateral DON exposure). Lgr5-EGFP^+^ cells (green) shows Lgr5^+^ stem cells. (**B**) Representative confocal images of enteroids at 24 h after treatments. EdU^+^ cells (red) show proliferative cells and nuclei stained by Hoechst 33342 (sky blue). (**C**) The ratio of Lgr5^+^ cell numbers in enteroids at 24 h/0 h after treatments. Mean ± SEM, *n* = 8–16. (**D**) EdU^+^ cell quantification in enteroids at 24 h after treatments. The number of EdU^+^ cells was normalized with the number of total cells and expressed as EdU/total cells (%). Mean ± SEM, *n* = 16–22. Different lowercase letters indicate significant differences (*p* < 0.05; Tukey’s post hoc test). Scale bars: 20 µm.

**Figure 5 toxins-12-00610-f005:**
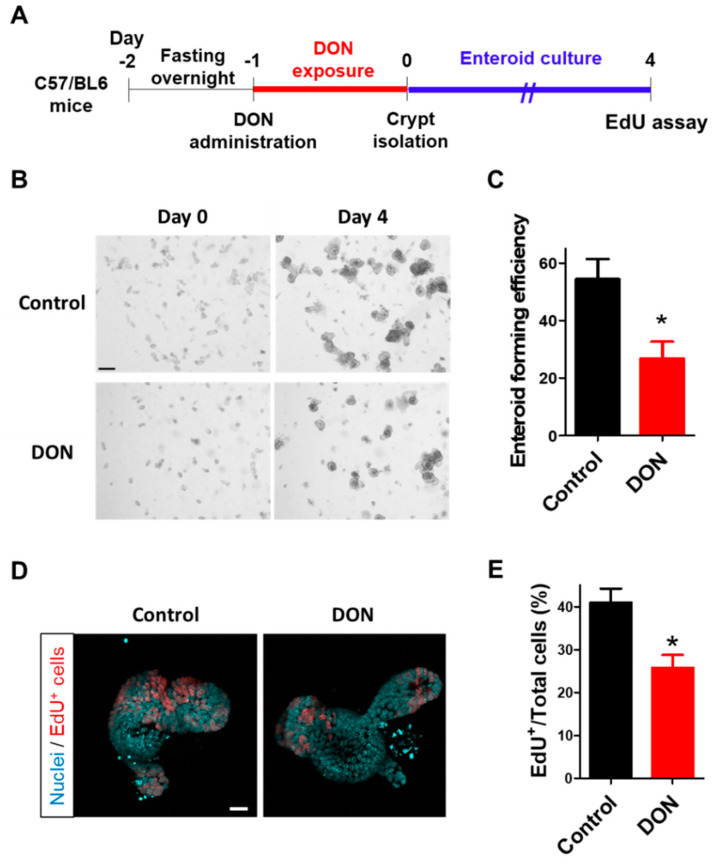
Effects of oral administration of DON to mice on intestinal stem cells. (**A**) Scheme of the experimental design. After fasting overnight, Wild-type (WT) mice were orally administered with DON at a dose of 50 mg/kg body weight. After the crypts were isolated from the mice at 24 h after DON exposure, enteroids were prepared and cultured for four days. (**B**) Representative images of enteroids (at day 0 or 4 after crypt isolation) derived from mice with or without oral DON administration. Scale bar: 200 µm. (**C**) Enteroid-forming efficiency from mice with or without oral DON administration. Enteroid-forming efficiency was calculated from the ratio of the number of enteroids at day 4 to the number of crypts at day 0. Mean ± SEM, *n* = 8–12. Asterisk (*) indicates a significant difference. (*p* < 0.05; Student′s *t*-test.) (**D**) Representative confocal images of enteroids at day 4 after crypt isolation. EdU^+^ cells (red) show proliferative cells and nuclei stained by Hoechst 33342 (sky blue). Scale bar: 20 µm. (**E**) EdU^+^ cell quantification in enteroids at day 4 after crypt isolation. The number of EdU^+^ cells was normalized with the number of total cells and expressed as EdU/Total cells (%). Mean ± SEM, *n* = 12–18. Asterisk (*) indicates a significant difference (*p* < 0.05; Students *t*-test).

**Figure 6 toxins-12-00610-f006:**
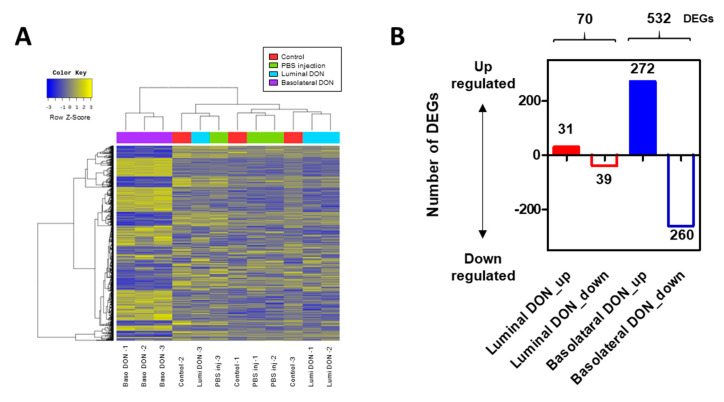
RNA sequencing (RNA-seq) analysis of enteroids after the basolateral or luminal DON exposure. (**A**) Heat map of 888 differentially expressed genes (DEGs) identified through comparison among control enteroids, PBS-injected enteroids, and enteroids after luminal or basolateral DON exposure. (**B**) Distribution of up- or downregulated DEGs in enteroids after luminal and basolateral DON treatment analyzed through RNA-seq analysis.

**Figure 7 toxins-12-00610-f007:**
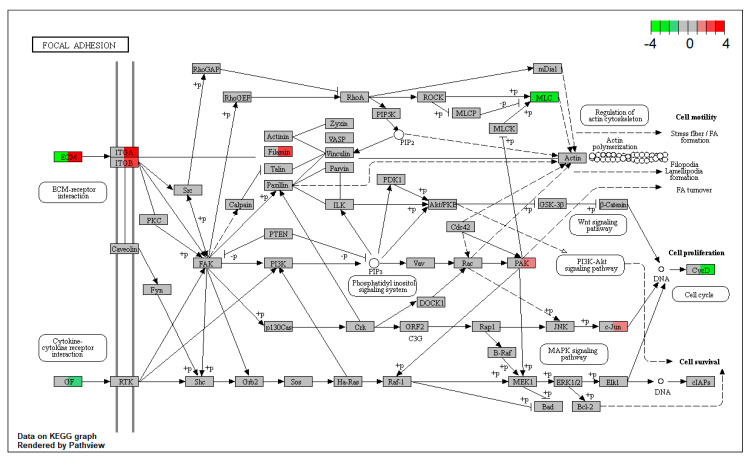
Kyoto Encyclopedia of Genes and Genomes (KEGG) pathway analysis of enteroids after basolateral or luminal DON exposure. Kyoto Encyclopedia of Genes and Genomes (KEGG) pathway of focal adhesion (mmu04510). Green- or red-colored gene name box indicates down- or upregulated genes in enteroids after luminal (left) or basolateral (right) DON exposures, compared with the control enteroids as log2 (fold change) of the expression value.

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
