# Peer review of "Mycotoxin Deoxynivalenol Has Different Impacts on Intestinal Barrier and Stem Cells by Its Route of Exposure"

_toxins, 2020, doi:10.3390/toxins12100610_

Round 1
Reviewer 1 Report
The manuscript is an excellent work that studies the toxicity of deoxynivalenol (DON) in intestinal cells by resorting to a novel and original approach based on intestinal organoids. The results achieved are of great significance since the authors prove that DON toxicity occurs mainly via basolateral exposure. I found only one point that needs to be revised by the authors. In page 15, the statistics should be added to Figure S3 C.
In the future, it would be interesting if authors could test DON derivatives (acetyl and glucoside forms and de-epoxy-DON) using the same method. It would be interesting to know whether or not they behave like DON.
Reviewer 2 Report
The authors present a well constructed and very interesting manuscript concerning study on a highly relevant topic to the toxic effects of DON specially on intestinal barrier function. The authors are to be commended for the scope and depth of their study, which has clearly represented much time and effort. Unfortunately I have some minor comments to the manuscript:
Line 62: “Our goal is…” it would be more appropriate “Our goal was …”
Line 65-66: “Here, we reported that basolateral DON exposure was more deleterious to intestinal barrier functions and stem cells than luminal DON exposure.” This sentence should be rather in conclusion.
Line 283: How many animals were used in experiment? How many male and female mice were used for experimentation? or How many enteroids were collected from one mouse, when was be one?
Line 285: Did feed be tested for DON and another mycotoxins?
Line 423: n = 4–7. This marking is not understandable; SEM is proper an abbreviation
Line 288: Which was the basis for administering DON at a dose of 50 mg/kg BW? Isn't that a gut-toxic level? Is this dose comparable to 1µM DON?
The literature should be organized in accordance with Instructions for Authors: e.g. Abbreviated Journal Name etc.!
There is no abbreviation in the manuscript.
I suggest its acceptance after minor corrections.
